# Applications of an Electrochemical Sensory Array Coupled with Chemometric Modeling for Electronic Cigarettes

**DOI:** 10.3390/s24175676

**Published:** 2024-08-31

**Authors:** Bryan Eng, Richard N. Dalby

**Affiliations:** School of Pharmacy, University of Maryland, Baltimore, MD 21201, USA; bryan.eng@umaryland.edu

**Keywords:** electronic cigarettes, nicotine, sensory array, menthol, vitamin E, chemometrics, volatile organic compounds (VOC), counterfeit detection, tobacco

## Abstract

This study investigates the application of an eNose (electrochemical sensory array) device as a rapid and cost-effective screening tool to detect increasingly prevalent counterfeit electronic cigarettes, and those to which potentially hazardous excipients such as vitamin E acetate (VEA) have been added, without the need to generate and test the aerosol such products are intended to emit. A portable, in-field screening tool would also allow government officials to swiftly identify adulterated electronic cigarette e-liquids containing illicit flavorings such as menthol. Our approach involved developing canonical discriminant analysis (CDA) models to differentiate formulation components, including e-liquid bases and nicotine, which the eNose accurately identified. Additionally, models were created using e-liquid bases adulterated with menthol and VEA. The eNose and CDA model correctly identified menthol-containing e-liquids in all instances but were only able to identify VEA in 66.6% of cases. To demonstrate the applicability of this model to a commercial product, a Virginia Tobacco JUUL product was adulterated with menthol and VEA. A CDA model was constructed and, when tested against the prediction set, it was able to identify samples adulterated with menthol 91.6% of the time and those containing VEA in 75% of attempts. To test the ability of this approach to distinguish commercial e-liquid brands, a model using six commercial products was generated and tested against randomized samples on the same day as model creation. The CDA model had a cross-validation of 91.7%. When randomized samples were presented to the model on different days, cross-validation fell to 41.7%, suggesting that interday variability was problematic. However, a subsequently developed support vector machine (SVM) identification algorithm was deployed, increasing the cross-validation to 84.7%. A prediction set was challenged against this model, yielding an accuracy of 94.4%. Altered Elf Bar and Hyde IQ formulations were used to simulate counterfeit products, and in all cases, the brand identification model did not classify these samples as their reference product. This study demonstrates the eNose’s capability to distinguish between various odors emitted from e-liquids, highlighting its potential to identify counterfeit and adulterated products in the field without the need to generate and test the aerosol emitted from an electronic cigarette.

## 1. Introduction

In the evolving landscape of nicotine delivery systems, electronic cigarettes (e-cigarettes) have emerged as a controversial yet prevalent alternative to traditional tobacco products. Initially conceived as a smoking cessation aid and a safer alternative to combustible cigarettes, e-cigarettes have gained immense popularity, particularly among younger demographics [1]. However, this burgeoning market has also given rise to an alarming trend: the proliferation of counterfeit e-cigarettes [2,3]. These illegitimate products undermine public health initiatives and pose significant health risks to consumers [2,3].

The importance of addressing the issue of counterfeit e-cigarettes cannot be overstated. We have seen the global impact of illicit trade in cigarettes and can expect similar lasting impacts from the trade of counterfeit e-cigarettes [4]. Firstly, these counterfeit products often ignore regulatory standards and prudent quality controls, leading to the possibility of more user exposure to hazardous materials such as lead, a constituent of inexpensive soldering joints, and inconsistent nicotine levels. This variability can exacerbate health risks, including nicotine addiction and respiratory disease. Furthermore, counterfeit e-cigarettes may become a conduit for potentially dangerous modifications and adulteration [3].

Secondly, the rise of counterfeit e-cigarettes undermines efforts of public health officials to monitor, evaluate and regulate the e-cigarette industry, and legitimate manufacturers seeking to improve the safety of their products are harmed when counterfeit products flood the market and reduce consumer confidence.

Lastly, counterfeit e-cigarettes create unfair market competition for legitimate manufacturers and distributers, potentially leading to financial and job losses, while simultaneously evading taxation [4]. One may be unsympathetic concerning the problems faced by regulated e-cigarette manufacturers while acknowledging that an underground market in counterfeits of their products poses even more risk to public health.

Currently, methods to detect counterfeit tobacco products rely on inductively coupled plasma-optical emission or mass spectrometry, gas chromatography, and X-ray fluorescent spectrometry, which all require extensive sample preparation and are generally unsuitable for use in the field [5].

This paper aims to provide a framework for the detection of counterfeit e-cigarettes using electronic nose sensor arrays coupled with chemometrics. Our assessment focused on the MSEM 160 e-Nose sensor, which was developed for detecting and sampling gases, volatile organic compounds (VOCs), and other airborne chemicals. It is compact and portable as well as consistent with field deployment for analyzing e-liquids. The MSEM 160 features targeted gas sensors for hydrogen sulfide, ammonia, and hydrocarbons along with a sensor array designed for VOCs and other odor-inducing compounds. We evaluated the system’s ability to classify e-cigarette brands, aiming for quicker results and lower costs compared to traditional analytical methods for identifying counterfeited and adulterated products.

### 1.1. Vitamin E Acetate

Vitamin E, a fat-soluble antioxidant, is an essential nutrient found naturally in various foods and used widely in dietary supplements and skincare products. While beneficial in these contexts, its derivative, vitamin E acetate (VEA), has emerged as a compound of significant concern when used in e-cigarette e-liquids, particularly those in illicit products. VEA is used as a thickening agent and seems particularly prevalent in e-liquids containing tetrahydrocannabinol (THC) [6].

When vaporized by the heating coil of an electronic cigarette and inhaled, thermal decomposition products of VEA have been linked to e-cigarette or vaping product use-associated lung injury (EVALI) [7]. Patients with EVALI present with respiratory symptoms ranging from cough and shortness of breath to severe lung injury requiring hospitalization.

### 1.2. Menthol

Menthol, a natural compound derived from mint oils, is used for its cooling and soothing properties in products, ranging from pharmaceuticals to confectioneries. In e-cigarettes, menthol is deployed in e-liquid formulations as a flavoring and cooling agent, but like VEA, heating may result in the formation of degradation products with an uncertain safety profile. Emerging scientific evidence suggests potential health hazards associated with menthol when inhaled through e-cigarettes, necessitating a critical examination of its implications [8]. The FDA has moved to ban menthol flavorings in several forms of tobacco products [3].

One of the primary concerns is the potential for menthol to enhance nicotine addiction. Studies suggest that menthol may modulate nicotine’s addictive properties, increasing the frequency of use [9]. The analgesic effects of menthol on nicotine irritation have also been demonstrated in humans [10]. This poses significant public health challenges, especially considering the popularity of menthol flavors among youth and their potential role as a gateway to sustained nicotine dependency.

Furthermore, the respiratory implications of inhaling mentholated e-liquids are not well understood. While menthol’s cooling effect may mask the harshness of nicotine and other chemicals, it may also lead to deeper inhalation and increased exposure to harmful constituents present in e-liquids [11]. Studies also suggest that menthol smoke induces more severe lung inflammation [12]. The long-term pulmonary effects of this increased exposure remain a critical area of concern.

### 1.3. Gas Sensors

The foundational principle of the electronic nose is based on the biological olfactory pathway, where the olfactory receptors in the nose bind to odor molecules and send signals to the brain for odor recognition and differentiation [13]. Similarly, an electronic nose (Figure 1) comprises an array of sensors designed to respond to various volatile organic compounds (VOCs) in the air [14,15]. The sensors, which can be made from a range of materials including polymers, metal oxides, and conducting polymers, generate a signal pattern from the detected VOCs [16]. This pattern, often referred to as an ‘odor fingerprint’, is then processed using algorithms and pattern recognition systems, akin to the way the brain processes smells, to identify, compare, or quantify various odors [17].

The sensors in an electronic nose can detect and analyze the unique chemical signatures in the headspace above the e-liquid. These signatures are influenced by the specific combinations of propylene glycol, glycerin, flavorings, and nicotine that manufacturers use. Counterfeits often use lower-quality or different proportions of these ingredients, leading to an altered chemical composition in the bulk liquid and vapor phase [3]. Illicit products may also contain undisclosed ingredients. By employing pattern recognition and machine learning techniques, the e-nose system can compare the chemical signature of a sample against the known profiles of authentic products. This study explored the ability of such an approach to distinguish between e-liquids of various compositions with a focus on the detection of potentially harmful or illegal additions and differentiating brand products from copies.

## 2. Materials and Methods

### 2.1. Sensory Array and Sampling Procedure

The portable environmental monitor MSEM 160 (Sensigent LLC, Baldwin Park, CA, USA) was utilized to detect and analyze the volatile substances emanating from e-liquids. This instrument is outfitted with 32 sensors, which include ones for measuring temperature and humidity (sensors 1–4). It includes a range of metal oxide semiconductor (MOS) sensors (sensors 5–12) designed for the electrochemical detection of specific chemicals and other electrochemical detection units tuned to detect hydrogen sulfide (sensor 13), ammonia (sensor 14), and hydrocarbons (sensor 15). A photoionization (PI) detector is intended to respond to all volatile organic compounds (total VOCs, sensor 16), and a series of polymer composite (PCSs) sensors (sensors 17–32) respond to other molecules in the vapor phase [18]. Sensor outputs were recorded every second over the sample period [18]. A feature of such sensor arrays is that sensors may respond to a specific chemical or a related one. While this lack of specificity may be a limitation in some applications, it does give the instrument the ability to output a range of signals (a pattern or fingerprint) in the absence of information about the composition of the vapor or gas that is being evaluated.

The instrument was turned on, and the sensors were allowed to equilibrate for at least 45 min before taking measurements. The sample apparatus features a closed-loop system which consisted of a Chemglass impinger (Vineland, NJ, USA) connected to the MSEM 160 eNose using a 1/8th inch internal diameter Tygon tubing and Luer-loc fittings, which were both obtained from McMaster-Carr (Elmhurst, IL, USA). Prior to use, impingers were tripled rinsed with soap and water, dried with clean compressed air, and then placed in a 60 °C incubator (MYTEMP 65, Benchmark Scientific Inc., Sayreville, NJ, USA) for at least 15 min. The MSEM 160 eNose sampling flow rate was set to 0.45 L per min (L/min).

#### 2.1.1. E-Liquid Components and Adulterants

To identify the components of the e-liquid and potential adulterants, four models were generated. Approximately 1.0 g of each e-liquid base was pipetted into a gas impinger. The headspace of the impinger was sampled for one minute per replicate with a one-minute pre- and post-purge cycle with ambient air to clean the gas path before and after sampling. 1, 2-Propanediol (PG), glycerol (G), and (-)-nicotine purchased from Sigma-Aldrich (St. Louis, MO, USA) were used in the e-liquid base creation. The first replicate of each sample set was used to prime the instrument, and was not included in the data analysis (Figure 2). Gas and vapor within the headspace was drawn into the internal chamber, where an array of 32 sensors had the opportunity to interact with the gas or vapor. The electrical output of each sensor, relative to the baseline, depends on the extent of interaction with molecules in the headspace.

##### Model 1: Propylene Glycol and Glycerol

The first model was generated by sampling six replicates each of PG and G. Six additional replicates of each were generated to assess the predictive rate for each group.

##### Model 2: Nicotine Addition

A commercially relevant mixture of PG and G (70:30 by weight) was created. To a portion of this mixture, 5% nicotine was added, resulting in two mixtures: one blank e-liquid and one containing nicotine. Six replicates of each mixture were sampled to generate the model. An additional six replicates of each were generated to assess the model’s predictive capabilities.

##### Models 3–5: Menthol and Vitamin E Acetate Adulteration

Using the 70:30 mixture of PG and G created previously, two separate mixtures were prepared: one containing 1% menthol and the other 10% VEA both purchased from Alfa Aesar (Lancashire, UK). Twelve replicates of each mixture were sampled using the method outlined above. Six additional replicates were generated to determine the predictive rate of the model.

##### Models 6–8: Menthol and Vitamin E Acetate Adulteration in JUUL

Using the commercial e-liquid extracted from JUUL (JUUL Labs Inc., San Francisco, CA, USA, Virginia tobacco flavor), two separate mixtures were prepared, one containing 1% menthol and the other 10% VEA by weight. Six replicates of each mixture were sampled using the method outlined above. Twelve additional samples were generated for the menthol mixture and nine were generated for VEA to assess the model’s predicative capability.

These models were used to distinguish between the different adulterants in the e-liquid samples.

#### 2.1.2. Commercial E-Liquids

Six brands of commercially available electronic cigarettes were studied (Table 1).

E-liquid samples were taken from e-cigarette brands purchased from retailers in the mid-Atlantic region of the United States. Popular brands were selected for study. An aliquot of approximately 20 mg of e-liquid was added to a gas impinger. The headspace of the impinger was sampled for one minute per replicate, with a one-minute pre- and post-purge cycle to clean the gas path before and after sampling. The first replicate of each sample set was used as a priming measurement and was not included in the data analysis. Each brand was sampled 18 times for model generation.

To assess interday variability, this study was repeated on two consecutive days, and all three days of sampling were compiled for model generation. Nine replicates of each brand were generated on another day to be used as a prediction set.

Using the e-liquid from Clear Elf Bar and Mystery Mix Hyde IQ, three alterations were made as detailed in Table 2. Three replicates of each alteration were sampled using the eNose method described previously. The data were added to a support vector machine (SVM) model previously used to identify brands.

### 2.2. Data Analysis

Files containing all sensor outputs generated from the MSEM 160 were loaded into the Chemometric Data Analysis software (Sensigent LLC, Baldwin Park, CA, USA). The sensor outputs were filtered with the Savitzky–Golay algorithm to reduce their signal-to-noise ratio [19,20] and subsequently converted into ΔR/R (change in response (R)/baseline response) before being loaded for model training [21]. Sensors 1–4 were deselected from the sensor selection menu, since they measure only temperature and humidity. Using supervised training techniques, the data were normalized, auto-scaled, and the canonical discriminant analysis (CDA) algorithm, a type of linear discriminant analysis (LDA), was applied [22]. The model was then cross-validated using the dataset used to generate the model. The method files were saved, and prediction sets were added to test validity. For the training sets based on six commercial e-liquids and their adulterated versions, the SVM algorithm was also applied [23].

## 3. Results

Ten models were generated. Details regarding the e-liquid types, algorithm, and cross-validation results can be found in Table 3.

### 3.1. E-Liquid Components

#### 3.1.1. Propylene Glycol and Glycerol

The first model sought to distinguish between propylene glycol (PG) and glycerol (G) (Figure 3). Six replicates of each e-liquid base were sampled, resulting in a total of 12 replicates. The eNose accurately differentiated between the two components, demonstrating its ability to identify the primary bases used in e-liquid formulations. The cross-validation accuracy for this model was 94.4%, and it achieved a 100% prediction rate using a validation set containing six samples of each.

#### 3.1.2. Nicotine Addition

The second model was designed to detect the presence of nicotine in a commercially relevant mixture of PG and G (70:30) (Figure 4). A portion of this mixture was spiked with 5% nicotine, and six replicates of each mixture (blank e-liquid and nicotine-containing e-liquid) were sampled. The eNose successfully identified the presence of nicotine, highlighting its capability to detect significant components in e-liquid formulations.

#### 3.1.3. Menthol and Vitamin E Acetate Adulteration

The third model aimed to identify adulterants, specifically menthol and VEA, in e-liquid bases (Figure 5). Using the 70:30 mixture of PG and G, two separate mixtures were prepared, containing 1% menthol and 10% VEA, respectively. Twelve replicates of each mixture were sampled. The eNose correctly identified menthol-containing e-liquids in all instances. However, the identification accuracy for vitamin E acetate was 66.6%, indicating a need for further refinement in detecting this specific adulterant.

#### 3.1.4. Menthol and Vitamin E Acetate Adulteration in JUUL

The fourth model aimed to identify adulterants, specifically menthol and VEA, in commercial e-liquids (Figure 6). Using Virginia Tobacco JUUL e-liquid, two separate mixtures were prepared, containing 1% menthol and 10% VEA, respectively. Six replicates of each mixture were sampled. The eNose correctly identified menthol-containing e-liquids 91.6% of the time and 75% of the time for those containing VEA.

### 3.2. Commercial Product Testing

The eNose device demonstrated a high level of accuracy in identifying electronic cigarette brands. Each brand was sampled 18 times for model generation with the first sample excluded from the analysis to avoid variability due to priming the sampling circuit. The data analysis involved filtering the sensor outputs with the Savitzky–Golay algorithm, converting the outputs into ΔR/R, and applying supervised training techniques.

To demonstrate the eNose’s ability to discriminate among commercial products, a model using six different e-liquids extracted from them was generated. Each brand was sampled 18 times, and CDA was applied to the dataset. This model yielded a cross-validation accuracy of 91.7%, but it only contained data generated on one day (Figure 7).

When the dataset was expanded to include data collected on three different days, with 18 samples of each brand per day (108 samples per day, totaling 324 samples), the cross-validation accuracy of the CDA model dropped to 41.7%.

A validation set of the six commercial brands was tested by generating nine samples from each brand, totaling 54 samples. The model correctly predicted 51 out of the 54 samples, resulting in a prediction accuracy of 94.4% (Figure 8).

To address this issue, the SVM algorithm was applied to the same dataset, which resulted in a substantial improvement in cross-validation accuracy to 84.7%. This indicates that SVM is more robust and can better account for differences in data collected on different days. These findings suggest that while CDA can be effective under controlled, single-day conditions, SVM provides a more reliable approach for models involving interday variability, enhancing the eNose’s practical applicability in real-world settings. These findings are in line with literature suggesting that SVM has high accuracy and tolerance for irrelevant or redundant data [13].

To simulate counterfeit products, actual product e-liquid formulation ratios were intentionally altered. The electronic cigarette brand identification model was able to discriminate these adulterated formulations from the unaltered reference product in all cases [3] (Figure 9, Table 4).

## 4. Discussion

This study investigated the use of an eNose device as a screening tool for identifying counterfeit and adulterated electronic cigarette e-liquids. The results demonstrate that the eNose can differentiate between various e-liquid components and detect specific adulterants, suggesting it may be of use in enforcement, regulatory and commercial settings.

Cross-validation was performed on each set of data to evaluate the performance and generalizability of predictive models before prediction datasets were added to validate their accuracy. This is important to mitigate overfitting, provide a robust performance estimate, and efficiently utilize limited data [24,25].

Following instrument equilibration, samples could be analyzed, and the instrument readied for reuse in approximately 4 min. Sample preparation was minimal compared to techniques more commonly used for counterfeit detection, and it was also amenable to field applications.

The eNose distinguished between propylene glycol (PG) and glycerol (G) and detected the addition of nicotine to e-liquid mixtures. These findings highlight the device’s capability to accurately identify primary components and significant additives in e-liquids under controlled conditions.

The eNose identified menthol-containing e-liquids reliably, but its identification accuracy for VEA was lower, indicating the need for further refinement in detecting this specific adulterant. The lower accuracy for VEA could be attributed to its less distinctive odor profile compared to menthol, suggesting that sensor sensitivity needs to be improved or additional sensors developed. Additionally, elevation of the sample chamber temperature could increase the vapor pressure of some analytes and result in stronger sensor responses.

The eNose demonstrated high accuracy in identifying electronic cigarette brands under controlled conditions. However, performance significantly declined when data from different days were included, highlighting the impact of interday variability. This could be attributed to drift in the performance of the sensors, which could be alleviated by a drift compensation algorithm such as domain adaptive extreme learning machines (DAELM) [26]. Model overfitting is unlikely as each model was cross-validated. The application of the support vector machine (SVM) algorithm substantially improved model reliability, making it a preferable choice for longitudinal studies and practical field applications.

The eNose’s performance seems to improve with more potent odors, as demonstrated in the comparisons between menthol, a substance with a relatively high vapor pressure and odor, and VEA, which is an essentially odorless, viscous substance with a low vapor pressure. The discrimination of potent odors may make the E-nose useful for detecting flavorings in e-liquids when taking a sniff may expose a human or animal to harm (for example, where the possibility of illicit drug addition exists).

Interday variability significantly affected the model’s performance when using CDA, highlighting the challenge of maintaining consistent sensor responses over time. The substantial drop in cross-validation accuracy with the inclusion of data from different days illustrates the importance of considering temporal factors in model development. The SVM algorithm’s ability to handle temporal variability more robustly suggests its suitability for practical applications, ensuring the reliability and consistency of the eNose.

## 5. Conclusions

This study demonstrates the eNose’s potential as a rapid and cost-effective screening tool for detecting counterfeit and adulterated e-liquids. The device’s ability to accurately detect specific e-liquid adulterants and differentiate between e-cigarette products may have applications in law enforcement, regulatory and commercial settings. However, substantial interday variability in predictability and the lower ability for detecting low odor additives indicate areas for further model and hardware refinement. Future research should focus on improving the eNose’s sensitivity to a broader range of adulterants, addressing temporal variability, and exploring advanced algorithms to enhance model robustness. Future studies could incorporate advanced pattern recognition algorithms such as linear ridge classifier (LRC), 3-order polynomial classifier (PO3), decision tree (DT), random forest (RF), and multilayer perceptron (MLP) [27]. These efforts should contribute to the development of reliable and effective screening tools for numerous applications, including maintaining the integrity of e-cigarette product supply chains for those adults who choose to use them, and the collection of data to support regulatory decision making to maximize public health.

## Figures and Tables

**Figure 1 sensors-24-05676-f001:**
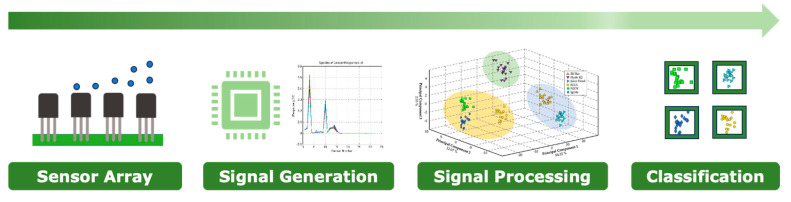
The process for applying of machine learning techniques to gas sensors responses for the classification for e-liquid brand classification.

**Figure 2 sensors-24-05676-f002:**
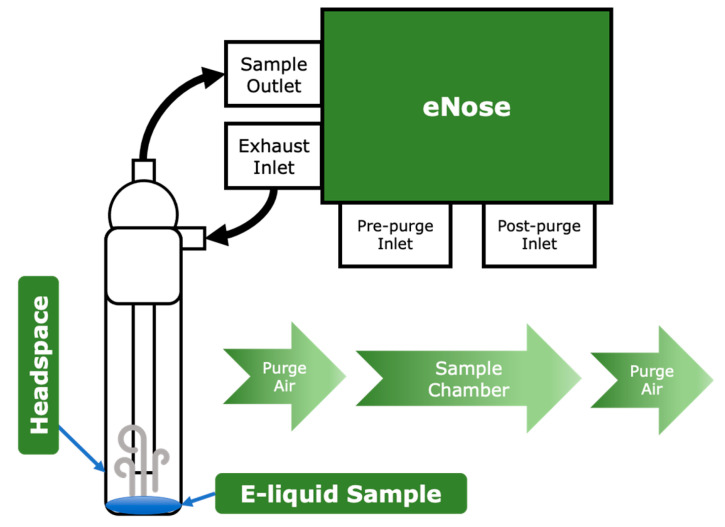
Schematic diagram of the headspace sampling apparatus and gas sampling procedure. Sampling occurs in a closed-loop system, and the internal eNose chamber containing the sensors is purged before and after sampling.

**Figure 3 sensors-24-05676-f003:**
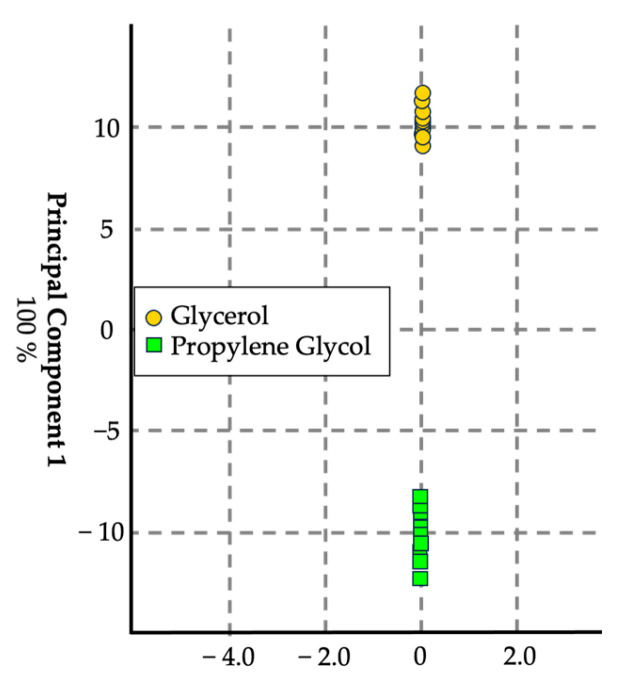
Score plot in canonical space with auto-scale generated using the chemometric data analysis software for visualization. *n* = 36, with 18 replicates per set. This canonical discriminant analysis model yielded a cross-validation of 94.4%.

**Figure 4 sensors-24-05676-f004:**
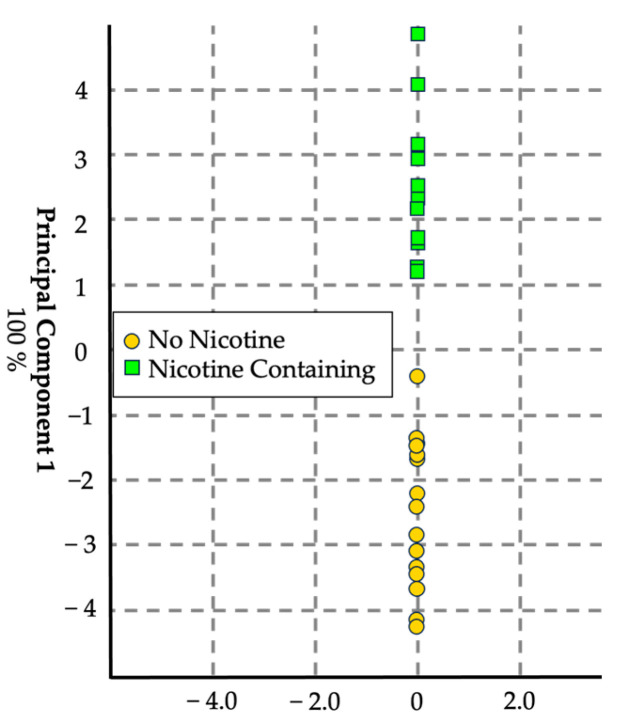
Score plot in canonical space with auto-scale generated using the chemometric data analysis software for visualization. *n* = 36, with 18 replicates per set. This canonical discriminant analysis model yielded a cross-validation of 94.4%.

**Figure 5 sensors-24-05676-f005:**
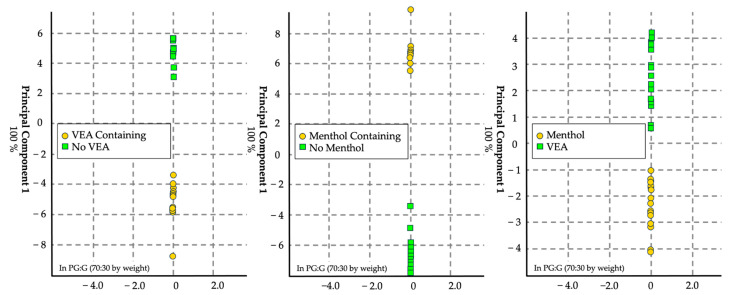
Score plot in canonical space with auto-scale generated using the chemometric data analysis software for visualization. *n* = 36, with 18 replicates per set. This canonical discriminant analysis model yielded a cross-validation of 94.4% and 97.2% for menthol and VEA, respectively, whereas the comparison between menthol and VEA yielded a 100% cross-validation.

**Figure 6 sensors-24-05676-f006:**
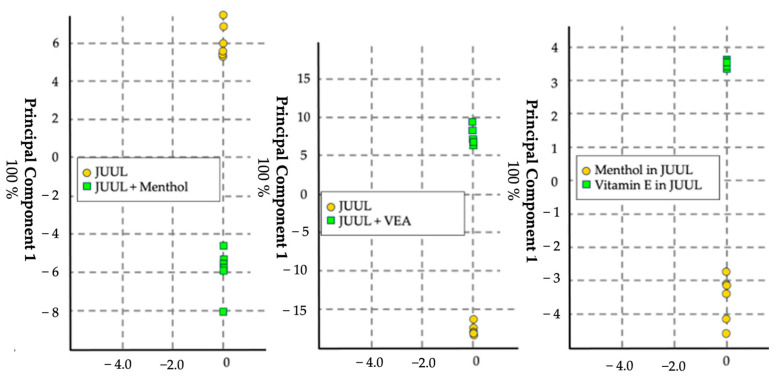
Score plot in canonical space with auto-scale generated using the chemometric data analysis software for visualization. *n* = 12, with 6 replicates per set. This CDA model yielded a cross-validation of 83.3% and 91.7% for VEA and menthol, respectively, whereas the comparison of menthol and VEA in JUUL yielded a cross-validation of 100%.

**Figure 7 sensors-24-05676-f007:**
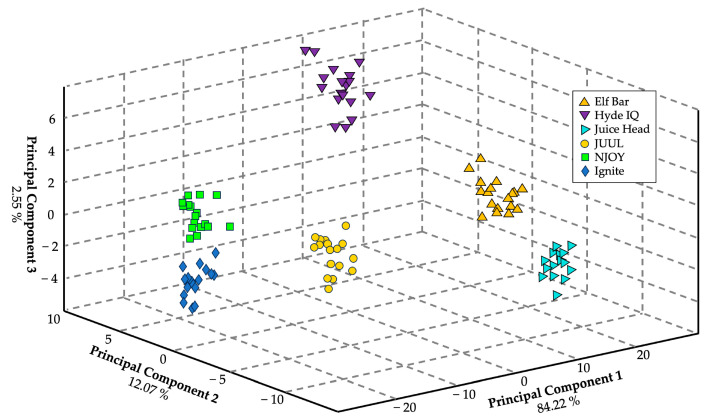
A three-dimensional CDA plot generated using the chemometric data analysis software for visualization. *n* = 108, with 18 replicates per brand. This CDA model yielded a cross-validation of 91.7%. The yellow highlighting indicates products with only tobacco flavoring, while blue represents those which also contain menthol, and green represents those with fruit flavorings.

**Figure 8 sensors-24-05676-f008:**
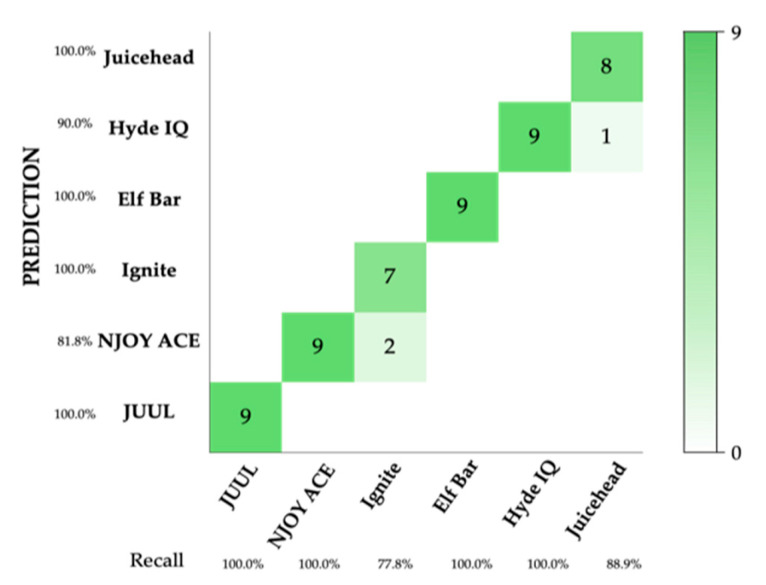
A predictive model was constructed using the SVM algorithm. *n* = 324. The model had an accuracy of 84.7% through cross-validation. Nine replicates of each brand were measured for a predication set. The figure above details the correct predictions for each of the brands in the form of a confusion matrix and yielded a total accuracy of 94.4%.

**Figure 9 sensors-24-05676-f009:**
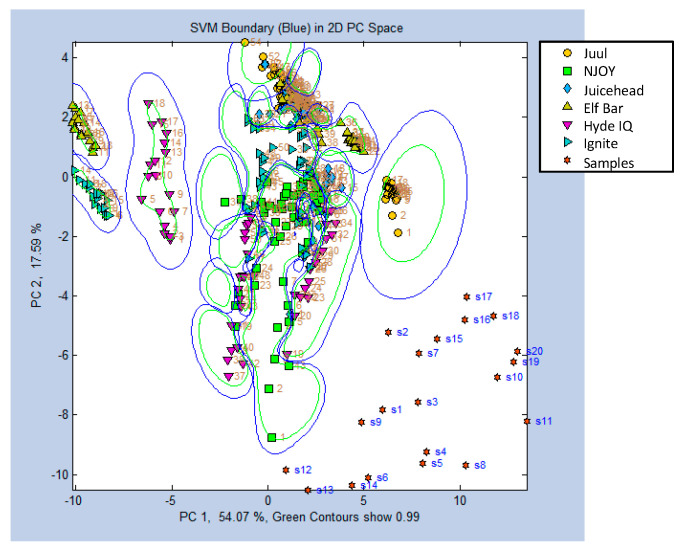
A two-dimensional SVM model was created and visualized using the chemometric data analysis software. *n* = 324, with 54 replicates per brand collected over three days (18 per day). Plotted along with the unchanged reference model data are altered sample (denoted as ‘s’) data for Elf Bar and Hyde IQ. The adulteration that was applied can be matched to the sample numbers using Table 4.

**Table 1 sensors-24-05676-t001:** Commercially available electronic cigarette products evaluated in this study.

Brand	Flavor	Manufacturer	Location	Batch Number
Elf Bar	Clear (Menthol)	Guang Dong QisiTech Co., Ltd.	Shenzhen, China	EP019125
Hyde IQ	Mystery Mix	Shenzhen IVPS Technology Co., Ltd.	Shenzhen, China	TT237802
Juice Head	Fresh Mint	MH Global LLC	Garden Grove, CA, USA	09292201
JUUL	Virginia Tobacco	JUUL Labs Inc.	San Francisco, CA, USA	HH15PA14A
NJOY Ace	Tobacco	NJOY LLC	Scottsdale, AZ, USA	2790105 2841
Ignite	Tobacco	Shenzhen VapeEZ Technology Ltd.	Shenzhen, China	0929/2021

**Table 2 sensors-24-05676-t002:** Adulteration of Clear Elf Bar and Mystery Mix Hyde IQ groups and ratios.

Brand	Flavor	Alteration and Ratio
		1:1 Addition of PG to the manufacturer’s e-liquid
Hyde IQ	Mystery Mix	1:1 Addition of PG to the manufacturer’s e-liquid
		Nicotine concentration increased to 24%
		1:1 Addition of PG to the manufacturer’s e-liquid
Elf Bar	Clear (Menthol)	1:1 Addition of PG to the manufacturer’s e-liquid
		Nicotine concentration increased to 24%

**Table 3 sensors-24-05676-t003:** Summary of each model generated and the corresponding cross-validation result.

Model	Samples Type	Matrix	Samples Per Set	Total Samples	Algorithm	CV
1	Glycerol	Not Applicable	18	36	CDA	94.4%
	Propylene Glycol	Not Applicable	18			
2	Nicotine	PG/G Mixture	18	36	CDA	94.4%
	No Nicotine	PG/G Mixture	18			
3	Menthol	PG/G Mixture	18	36	CDA	94.4%
	No Menthol	PG/G Mixture	18			
4	VEA	PG/G Mixture	18	36	CDA	97.2%
	No VEA	PG/G Mixture	18			
5	Menthol	PG/G Mixture	18	36	CDA	100.0%
	VEA	PG/G Mixture	18			
6	Menthol	JUUL	6	12	CDA	83.3%
	No Menthol	JUUL	6			
7	VEA	JUUL	6	12	CDA	91.7%
	No VEA	JUUL	6			
8	Menthol	JUUL	6	12	CDA	100.0%
	VEA	JUUL	6			
9	JUUL	Not Applicable	18	108	CDA	91.7%
	NJOY ACE	Not Applicable	18			
	Ignite	Not Applicable	18			
	Elf Bar	Not Applicable	18			
	Hyde IQ	Not Applicable	18			
	Juicehead	Not Applicable	18			
10	JUUL	Not Applicable	54	324	SVM	84.7%
	NJOY ACE	Not Applicable	54			
	Ignite	Not Applicable	54			
	Elf Bar	Not Applicable	54			
	Hyde IQ	Not Applicable	54			
	Juice Head	Not Applicable	54			

**Table 4 sensors-24-05676-t004:** Sample alterations and sample number key. Samples 1 and 11 were priming samples (not included in the prediction set) and thus are not included in the table.

Brands	Alteration	Sample Number
Hyde IQ	1:1 Addition of PG to the manufacturer’s e-liquid	2–4
	1:1 Addition of G to the manufacturer’s e-liquid	5–7
	Nicotine concentration increased to 24%	8–10
Elf Bar	1:1 Addition of PG to the manufacturer’s e-liquid	12–14
	1:1 Addition of G to the manufacturer’s e-liquid	15–17
	Nicotine concentration increased to 24%	18–20

## Data Availability

Data are contained within the article.

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
