# Peer review of "Applications of an Electrochemical Sensory Array Coupled with Chemometric Modeling for Electronic Cigarettes"

_sensors, 2024, doi:10.3390/s24175676_

Round 1

Reviewer 1 Report

Comments and Suggestions for Authors

Review Comments

This manuscript presents a compelling investigation into the application of eNose as a novel, portable tool for detecting counterfeit and adulterated electronic cigarettes. The study is timely and relevant given the increasing prevalence of counterfeit e-cigarettes, which pose significant public health risks. The authors provide a thorough exploration of the eNose's capabilities, supported by detailed experimental data. The use of an eNose for detecting counterfeit e-cigarettes is a novel application that could have significant implications for public health and regulatory enforcement. The study is well-designed, with two models tested to assess the eNose's accuracy in identifying various e-liquid components and adulterants. The findings suggest that eNose could be a valuable tool in the field for quickly identifying potentially hazardous e-cigarette products, which is a significant advantage over more time-consuming traditional methods. However, while the study offers great potential, several key issues need to be addressed to strengthen the study and its applicability. Therefore, I recommend a major revision of this manuscript before it is considered for publication.

The detailed comments are enumerated as follows:

1. In the introduction section, this manuscript only emphasizes the importance of addressing the problem of counterfeit e-cigarettes and does not present state-of-the-art research on identifying counterfeit products, i.e. traditional analytical identification methods. The goal of this research is to construct counterfeit product detection methods that are more efficient and less costly compared to traditional detection methods, but in the discussion section of this manuscript, there is no comparison with the accuracy and efficiency of traditional detection methods. Therefore, I suggest the authors add content in the introduction and discussion section to make the line more logical.

2. This study focused on menthol and vitamin E acetate (VEA), the main adulterants in e-cigarette liquids. The focus on these two specific substances is critical given their known health risks and prevalence in the marketplace, but the scope of the study would have been greatly enhanced by expanding the investigation to include more common adulterants and contaminants. E-cigarette liquids are often subject to various forms of adulteration, including the addition of harmful chemicals, synthetic cannabinoids, and other illicit substances that can pose serious health risks to consumers. By expanding the range of adulterants investigated, this study will not only provide a more comprehensive assessment of eNose’s functionality but also improve the overall reliability and utility of eNose as a screening tool.

3. Besides, I suggest the authors conduct an investigation or discussion about the concentrations of the test samples. The concentrations of the target atmospheres may have a strong impact on the response value of the gas sensors, so I wonder how the e-nose works when there are different concentrations of the same sample diluted by the clean air. The division of the data set in this manuscript can be illustrated using a figure, e.g., how many samples are in total, how to divide the training\validation\test set, and how to conduct the k-fold cross-validation method and random test. I also recommend the authors consider providing more visual figures such as the confusion matrix, the ROC curve, the Precision-Recall curve, and so on, with which the classification models can be evaluated more comprehensively.

4. The lower accuracy of eNose in detecting vitamin E acetate (VEA) compared to menthol suggests that eNose has limited detection capabilities and may have difficulty detecting low-odor adulterants. This may limit its effectiveness in identifying certain harmful substances in e-cigarette liquids. Furthermore, the temporal variability is a significant issue for the real application. Why did the accuracy decrease dramatically when randomized samples were presented to the model on different days? Is this due to overfitting of the model, drift in the performance of the sensor (influence of temperature and humidity), or instability of the experiment? The authors need to provide more in-depth discussion and analysis.

5. Currently, deep learning models and multi-task techniques are widely used and may be considered for comparison with the CDA and SVM models used in the manuscript. There are many studies about the deep learning methods of the electronic nose for VOCs detection. I suggest the authors have some review or discussion of these articles if appropriate.

[1] T. Wang, H. Zhang, Y. Wu, W. Jiang, X. Chen, M. Zeng, J. Yang, Y. Su, N. Hu, Z. Yang, Target discrimination, concentration prediction, and status judgment of electronic nose system based on large-scale measurement and multi-task deep learning, Sens. Actuators B: Chem. 351 (2022) 130915. https://doi.org/10.1016/j.snb.2021.130915.

Author Response

We thank both reviewers for their careful reading of the manuscript and their suggestions. Almost all suggestions were incorporated into the revised version of the manuscript, which we believe is now more comprehensive and easier to read and compehend. Thank you again for helping us improve the manuscript.

Reviewer 1

Comment 1: [In the introduction section, this manuscript only emphasizes the importance of addressing the problem of counterfeit e-cigarettes and does not present state-of-the-art research on identifying counterfeit products, i.e. traditional analytical identification methods. The goal of this research is to construct counterfeit product detection methods that are more efficient and less costly compared to traditional detection methods, but in the discussion section of this manuscript, there is no comparison with the accuracy and efficiency of traditional detection methods. Therefore, I suggest the authors add content in the introduction and discussion section to make the line more logical.]

Response 1: We agree with this observation so have mentioned currently employed detection methods in the introduction and discussion. Specifically:

Page 2, Introduction: “Currently, the most employed methods for counterfeit detection of tobacco include inductively coupled plasma-optical emission or mass spectrometry, gas chromatography, and X-ray fluorescent spectrometry which all require extensive sample preparation and cannot be used in the field.”

Page 13, Discussion: “Following instrument equilibration, samples could be analyzed, and the instrument readied for reuse in approximately 4 minutes. Sample preparation was minimal compared to techniques more commonly used for counterfeit detection, and amenable to field applications.”

Our work was intended to provide proof-of-concept utilizing a relatively new technology and not directly compare the accuracy and efficiency of eNose to existing methods. This may come later but is beyond the scope of the current manuscript.

Comment 2: [This study focused on menthol and vitamin E acetate (VEA), the main adulterants in e-cigarette liquids. The focus on these two specific substances is critical given their known health risks and prevalence in the marketplace, but the scope of the study would have been greatly enhanced by expanding the investigation to include more common adulterants and contaminants. E-cigarette liquids are often subject to various forms of adulteration, including the addition of harmful chemicals, synthetic cannabinoids, and other illicit substances that can pose serious health risks to consumers. By expanding the range of adulterants investigated, this study will not only provide a more comprehensive assessment of eNose’s functionality but also improve the overall reliability and utility of eNose as a screening tool.]

Response 2: We agree that our study would be enhanced by including other common and potentially more harmful adulterants. However, we do not possess the appropriate licenses or secure facilities to legally possess scheduled and illicit drugs. Vitamin E acetate and menthol were therefore chosen as relevant examples in the current public health and regulatory landscape. In addition, they represent low and high odor examples of adulterants, which helps the current proof-of-concept study show which other additives are likely to be readily detectable (those with a high odor and vapor pressure) and those which might be more challenging. We do not dispute the value of studying other harmful and potentially harmful drugs that have been or might be, added to e-liquids, but submit that it is beyond the scope of the current manuscript.

Comment 3: [Besides, I suggest the authors conduct an investigation or discussion about the concentrations of the test samples. The concentrations of the target atmospheres may have a strong impact on the response value of the gas sensors, so I wonder how the e-nose works when there are different concentrations of the same sample diluted by the clean air. The division of the data set in this manuscript can be illustrated using a figure, e.g., how many samples are in total, how to divide the training\validation\test set, and how to conduct the k-fold cross-validation method and random test. I also recommend the authors consider providing more visual figures such as the confusion matrix, the ROC curve, the Precision-Recall curve, and so on, with which the classification models can be evaluated more comprehensively.]

Response 3:

Two tables were added to illustrate the size and division of the data sets.

A confusion matrix was also added for the classifications of the six commercial brands.

The sensor responses do not change based on the volume of sample if the sample volume contains sufficient volatile component to saturate or establish equilibrium in the headspace and the closed sampling loop. This does not directly address the reviewer’s desire to explore different concentrations, but work to do that is ongoing. So far, we know that eNose can distinguish between typical (2.5 %) and reasonably foreseeable high concentrations (10 %) of nicotine, but we are uncertain about its ability distinguish between similar concentrations of nicotine. Unfortunately, this work will not be completed before the deadline for responding to reviewer comments, for which we sincerely apologize.

Comment 4: [The lower accuracy of eNose in detecting vitamin E acetate (VEA) compared to menthol suggests that eNose has limited detection capabilities and may have difficulty detecting low-odor adulterants. This may limit its effectiveness in identifying certain harmful substances in e-cigarette liquids. Furthermore, the temporal variability is a significant issue for the real application. Why did the accuracy decrease dramatically when randomized samples were presented to the model on different days? Is this due to overfitting of the model, drift in the performance of the sensor (influence of temperature and humidity), or instability of the experiment? The authors need to provide more in-depth discussion and analysis.]

Response 4:  We agree with the reviewers observation and have expanded the discussion on page 13.

“The eNose demonstrated high accuracy in identifying electronic cigarette brands under controlled conditions. However, performance declined when data from different days were included, highlighting the impact of interday variability. This could be attributed to drift in the performance of the sensors which could be alleviated by a drift compensation algorithm such as domain adaptive extreme learning machines (DAELM). Additionally, model overfitting is unlikely as each model was cross validated. The application of the support vector machine (SVM) algorithm substantially improved model reliability, making it a preferable choice for longitudinal studies and practical field applications.”

Comment 5: [Currently, deep learning models and multi-task techniques are widely used and may be considered for comparison with the CDA and SVM models used in the manuscript. There are many studies about the deep learning methods of the electronic nose for VOCs detection. I suggest the authors have some review or discussion of these articles if appropriate.]

Response 5: This paper was incredibly informative, and we agree that we should explore other models and task techniques. We have added a small section on page 13 and included the reference, but cannot apply these models to our data in the current manuscript.

“Future studies could incorporate advanced pattern recognition algorithms such as linear ridge classifier (LRC), 3-orderpolynomial classifier (PO3), decision tree (DT), random forest (RF), and multilayer perceptron (MLP).”

Reviewer 2 Report

Comments and Suggestions for Authors

Journal: Sensors

Manuscript ID: sensors-3169298

Title: Applications of an electrochemical sensory array coupled with chemometric modeling for electronic cigarettes

General Comments

This manuscript addressed the chemometric characterization of electronic cigarettes based on a electrochemical sensing technique with e-Nose. The analysis results is of interest mainly in the field of biochemistry and forensic analytical chemistry. However, I found some of the explanations difficult to follow, I suspect a reader less familiar with the topic might have even more significant difficulties. Hence I would suggest a minor revision of the manuscript. I annotate the manuscript with several corrections listed below.

1: Major comments

(1)                As above mentioned general comments, the results of this study are important, multivariate analysis are effective technique. However, I was unable to follow the direct detection results from the e-Nose system. The e-Nose is commercially available, but readers who are not familiar with it probably won't undetstand what kind of graphs are obtained from the e-Nose system. I thought that the paper would be more reader-friendly if you explained the primary data (graphs) from the e-Nose while showing representative examples.

(2)                In the case of the word "electrochemical measurement", it is often used to measure reaction systems in solutions. In a method for analyzing gases such as the e-Nose, it would be more reader-friendly to explain the principle of the measurement system. In other words, the detection principle should be briefly explained for the eNose part of Figure 1.

2: Minor comment

Please rotate the vertical axis of the figure 180 degrees.

Author Response

We thank both reviewers for their careful reading of the manuscript and their suggestions. Almost all suggestions were incorporated into the revised version of the manuscript, which we believe is now more comprehensive and easier to read and compehend. Thank you again for helping us improve the manuscript.

Comment 1: [As above mentioned general comments, the results of this study are important, multivariate analysis are effective technique. However, I was unable to follow the direct detection results from the e-Nose system. The e-Nose is commercially available, but readers who are not familiar with it probably won't understand what kind of graphs are obtained from the e-Nose system. I thought that the paper would be more reader-friendly if you explained the primary data (graphs) from the e-Nose while showing representative examples.]

Response 1: On page 2 we have added Figure 1 to visually explain the basic idea behind eNose. We hope this makes the paper more accessible to readers less familiar with sensor arrays and how they are utilized.  Thank you for the suggestion.

Comment 2: [In the case of the word "electrochemical measurement", it is often used to measure reaction systems in solutions. In a method for analyzing gases such as the e-Nose, it would be more reader-friendly to explain the principle of the measurement system. In other words, the detection principle should be briefly explained for the eNose part of Figure 1.]

Response 2: We completely agree. The new Figure 1 should help, and we have revised the text on page 4.

“Gas and vapor within the headspace was drawn into the internal chamber where an array of 32 sensors had the opportunity to interact with the gas or vapor. The electrical output of each sensor, relative to baseline, depends on the extent of interaction with molecules in the headspace.”

Round 2

Reviewer 1 Report

Comments and Suggestions for Authors

I believe that all my concerns from the previous round have been adequately addressed. The authors have successfully clarified the issues and enhanced the overall quality of the manuscript. The revisions have significantly improved the clarity, depth, and rigor of the research. Consequently, I recommend accepting the manuscript for publication.